# HP$^3$-NS: Hybrid Perovskite Property Prediction Using Nested Subgraph

## Abstract

Many machine learning techniques have demonstrated superiority in large-scale material screening, enabling rapid and accurate estimation of material properties. However, data representation on hybrid organic-inorganic (HOI) crystalline materials poses a distinct challenge due to their intricate nature. Current graph-based representations often struggle to effectively capture the nuanced interactions between organic and inorganic components. Furthermore, these methods typically rely on detailed structural information that hinders the applications of the methods for novel material discovery. To address these, we propose a nested graph representation HP$^3$-NS (Hybrid Perovskite Property Prediction Using Nested Subgraph) that hierarchically encodes the distinct interactions within hybrid crystals. Our encoding scheme incorporates both intra- and inter-molecular interactions and distinguishes between the organic and inorganic components. This hierarchical representation also removes the dependence on detailed structural data, enabling the model application to newly designed materials. We demonstrate the effectiveness and significance of the method on hybrid perovskite datasets, wherein the proposed HP$^3$-NS achieves significant accuracy improvement compared to current state-of-the-art techniques for hybrid material property prediction tasks. Our method shows promising potential to accelerate hybrid perovskite development by enabling effective computational screening and analysis of HOI crystals.

## 1 Introduction

Like many other fields, material science has seen a surge in the application of machine learning techniques to solve several problems and improve over traditional approaches. Nevertheless, most of these widely employed solutions use classical machine learning methods which learn on raw or manually pre-processed features (Cao et al., 2019; Li et al., 2019b; Ward et al., 2016) which can be challenging to optimize and get accurate performance as the number of features and the feature relationship complexity increases. Several works (Schutt et al., 2018; Tshitoyan et al., 2019) have tried to enhance the data representation by employing deep-learning-based data representation methods.

Recently, building upon data representation learning approaches, graph neural network (GNN) based methods (Xie & Grossman, 2018a; Choudhary & DeCost, 2021a) have shown to achieve state-of-the-art results in material prediction and discovery as they can learn a representation that models the intricate property relationships of the different atoms that make up the material compound. Nevertheless, these models assume the availability of the crystal structure of a material computed using density functional theory (DFT) methods (Jones, 2015). The practical application of these works is hindered since crystal structure data is not pre-available for every possible material. Consequently, utilizing the model for large-scale material screening and the discovery of new materials becomes extremely challenging, as it necessitates computing the Density Functional Theory (DFT) for each potential candidate. Several studies (Goodall & Lee, 2020b; Wang et al., 2021a; Schmidt et al., 2021) have addressed this issue by developing structure-agnostic models that yield comparable results.

These existing works have demonstrated promising results on materials exclusively composed of inorganic atoms, owing to the extensive availability of benchmark data. However, the application of graph neural networks to materials comprising both organic molecules and inorganic atoms—which

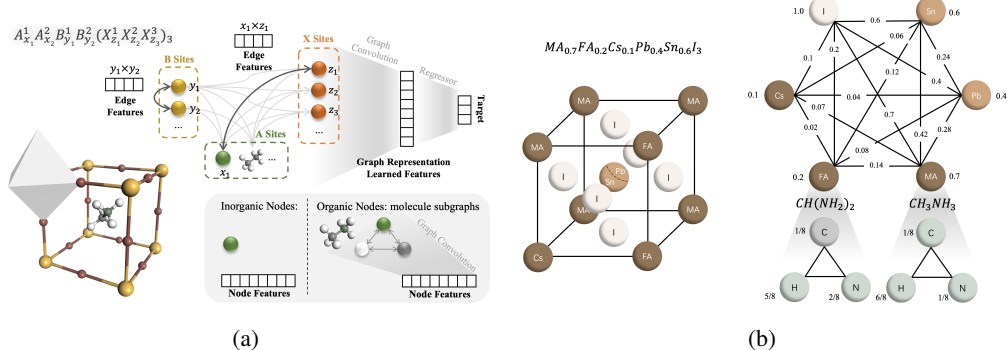

Figure 1: (a) HP³-NS: graph construction and representation learning process for hybrid organic-inorganic perovskites. (b) An example of how the graph is built out of the mixed perovskite's chemical composition.

have significant practical value in applications such as the design of alloys with desirable properties (Du et al., 2001; Nashrah et al., 2021), and energy-efficient perovskites (Kojima et al., 2009; Kumar & Naidu, 2021)—remains unexplored. In this study, we construct a nested graph neural network architecture to learn a graph representation that accommodates materials with mixed organic molecules and inorganic atoms. Furthermore, to reflect and encode the representation of atoms and molecules within a compound's crystalline structure, we have introduced an edge encoding mechanism employed to weigh the node features in the graph neural network architecture. Consequently, our proposed method accurately models mixed hybrid organic and inorganic compounds, surpassing the performance of commonly used classical methods and a graph neural network approach that employs a basic molecule feature representation.

Primarily, our contribution involves: (i) graph representation learning of hybrid organic and inorganic material compounds by constructing a nested graph representation that concurrently captures both molecule and compound representations; (ii) removing the need for structure data by emulating structural information within the edge encoding; (iii) demonstrating the effectiveness of the proposed method through empirical validation and experimental verification. The empirical experiments are conducted on a hybrid organic and inorganic perovskites (HOIPs) dataset, which has been curated and extracted from an open-source database. This lead us to name the proposed approach, Hybrid Perovskite Property Prediction Using Nested Subgraph (HP³-NS). Although HOIPs are used here to serve as a representative example for hybrid organic and inorganic material compounds due to their relatively larger experimental data, our model is versatile and can be directly applied to any hybrid organic and inorganic data. The general overview of our approach is depicted in Figure 1(a).

## 2 RELATED WORKS

**Graph Neural Networks for Materials:** graph neural networks (GNNs) models have come out as effective representation learning methods for data that can be represented as a graph. They learn a representation by passing and aggregating features to and from their adjacent nodes (Kipf & Welling, 2016; Veličković et al., 2017; Schlichtkrull et al., 2018). Building upon the achievements of GNNs in various domains, Xie & Grossman (2018b) presented Crystal Graph Convolutional Neural Networks (CGCNN) that integrates crystalline materials' structural and atomic properties by formulating a crystal graph and learning a graph representation that outperformed prior works. This shows the effectiveness of graph representation learning for material prediction and analysis. Subsequent to the development, various research works have been proposed to obtain improved performances. MEGNet (Chen et al., 2019), iCGCNN (Park & Wolverton, 2020), and TGNN (Na et al., 2020) have improved the CGCNN structure to attain significant advancements in both representation and generalization abilities, surpassing the initial CGCNN model and achieving state-of-the-art performance in predicting various material properties. ALIGNN (Choudhary & DeCost, 2021b), another notable GNN method for material prediction, proposed an approach that uses additional structural information, bond angle, to create a line graph in addition to the typical atomistic graph. This hi-

erarchical construction allows for accurate modeling of materials' atomic structure and chemistry, thereby enhancing the model's performance on various prediction tasks. Nevertheless, the practical application of these works is limiting as they require the material's structure to predict its properties, making them unsuitable for large-scale screening.

**Structure-Agnostic Material Property Prediction:** To remove the necessity for structure data, several works have proposed approaches that model materials solely using combinational and atomic features. Goodall & Lee (2020b) utilized material stoichiometry and atomic features in conjunction with a graph attention network(Veličković et al., 2018) to achieve a performance comparable to the structure-based approaches. In Goodall et al. (2022), they further improved the performance by adding a structure-related input that doesn't require DFT computation. Wang et al. (2021b) trains a transformer network using atoms as tokens and fractional features as positional encodings. Another work proposed in Schmidt et al. (2021) avoids the need for computed structure data by modeling the bond distance through embedding learning. Moreover, all material property prediction approaches (Li et al., 2019b; Cao et al., 2019) that use classical machine learning approaches only on material's atomic and stoichiometry data such as Ward et al. (2016) are also considered as structure-agnostic material prediction models. Our work aligns with these works as we also replaced structure data input by employing an edge encoding that mimics the structural positioning of the atoms or molecules in the material's crystal structure, aiming to apply our model for analyzing large-scale materials and discovering novel materials.

**Representation of Hybrid Organic-Inorganic Materials:** To the best of our knowledge, there has not been any prior research exploring the utilization of graph neural networks on hybrid organic and inorganic materials to separately learn representations for organic molecules, which are composed of atoms on their own. Usually, a simplistic approach is adopted where molecular features are represented by the weighted average features of its constituent atoms.

## 3 METHOD

In this section, we first highlight the graph neural network overview, and then we illustrate the details of our proposed method, including problem formulation, graph construction, and model architecture.

### 3.1 GRAPH NEURAL NETWORKS

Graph neural networks are neural network algorithms designed to operate on graph-structured data to learn node, relation, or graph representations. A graph is denoted by $G = (V, E)$ where $V = \{v_1, v_2, \ldots, v_M\}$ is the set of $M$ nodes and $E$ is the set of edges with $e_{ij}$ representing the weights of an edge that links node $i$ to node $j$. Each node can be associated with a node feature represented as $h_i \in \mathbb{R}^D$, a $D$ dimensional feature of $i^{th}$ node.

In graph representation, given a set of graphs $\mathcal{G} = \{G_1, G_2, \ldots, G_N\}$ and their corresponding labels = $\{y_1, y_2, \ldots, y_N\}$, graph neural network $g$ learns a graph representation $h_{G_m}$ of the graph $G_m$ using two operations, message(feature) passing, and read-out. message passing, $MP$, learns node representation by iteratively passing and aggregating a message between neighbor nodes,

$$\hat{h}_i^{\,k} = MP(h_i, h_j), \forall j \in \mathcal{N}_i \tag{1}$$

where $\hat{h}_i^{\,k}$ is the updated feature representation of node $i$ at the $k^{th}$ iteration and $\mathcal{N}_i$ is the set of nodes that are directly adjacent to node $i$.

On the other hand, readout operation, $RO$ is applied after $K$ iterations of message passing to combine the node features into a graph representation.

$$h_{G_m} = RO(\{h_i : \forall i \in G_m\}) \tag{2}$$

A regression/classification head network, $f$, is added on the top of the readout layer to map the graph representation into a continuous output prediction or a discrete classification, $\hat{y_m}$.

$$\hat{y_m} = f(h_{G_m}) \tag{3}$$

Usually, GNN architectures differ in how they define the message passing, read-out, and head network layers.

## 3.2 PROBLEM FORMULATION

A hybrid organic and inorganic compound can be represented as any mixture of atoms and molecules, $\mathcal{M}$,

$$\mathcal{M} = X^1_{r_{X1}} X^2_{r_{X2}} \dots X^N_{r_{XN}} \tag{4}$$

where $X^i$ can be any inorganic atom or an organic molecule with $r_{x^i}$ stochiometry ratio, and $N$ is the number of atoms and molecules that constitute the material.

The materials examined in our evaluation dataset are hybrid organic and inorganic perovskites (HOIPs), which are specific instances of hybrid organic-inorganic material. HOIPs are commonly denoted as $ABX_3$, a representation that carries a particular significance as it dictates the types of atoms or molecules that can occupy each of the $A$, $B$, and $X$ sites(positions), with their respective stoichiometry ratios being $1:1:3$. The $A$ site is the place where both organic molecules and inorganic atoms can occupy, while the other two sites are only occupied using inorganic atoms from different chemical groups. Each of the $A$, $B$, and $X$ can be combinations of multiple atoms/molecules, hence, we generally represent HOIPs as:

$$A^1_{r_{A1}} A^2_{r_{A2}} \dots A^N_{r_{AN}} B^1_{r_{B1}} B^2_{r_{B2}} \dots B^N_{r_{B3}} X^1_{r_{X1}} X^2_{r_{X2}} \dots X^N_{r_{XN}} \tag{5}$$

This representation will be used in the following sections.

## 3.3 GRAPH CONSTRUCTION

We framed our material prediction problem as a structure-agnostic graph representation learning that removes the necessity of crystal structure in building input graphs, rather seeks to construct a graph purely from the chemical formula of a material. the graph represents the material's structure, where nodes correspond to the atoms/molecules located at the A, B, or X sites, and edges indicate the frequency of interaction between neighboring nodes, mimicking how atoms and molecules are represented in the material's crystalline structure. Similarly, we can generate molecule subgraphs for organic nodes within a perovskite graph. The details of these constructions are detailed in the following subsections. An illustrative example of how the graphs are constructed can be found in Figure 1.

### 3.3.1 NODE DESIGN

Considering a node can be either an organic molecule or an inorganic atom, it is reasonable to classify the nodes of a graph into two groups: those representing inorganic atoms and those representing organic molecules, so we can process and apply transformation separately. For inorganic nodes, a feature vector is assigned to each of them, composed of fractional ratios of the atom represented by the node and eight atomic features obtained from a general-purpose descriptor set Magpie, list of the atomic features can be found in Appendix C. The node features will be non-uniformly binned and converted to categorical features for the benefit of smooth training. The details of the non-uniform binning can be found in Appendix B.

### 3.3.2 NESTED SUB-GRAPH CONSTRUCTION

Organic molecules in the A-site of a HOIP material contain organic molecules such as MA with the chemical formula $CH_3NH_3$, which, on their own, are composed of inorganic atoms. Hence, there needs to be a mechanism to generate a molecular representation that takes into account molecular composition and the relationship between the different constituent atoms. In previous works, the node features for organic sites are normally generated by statistically merging all features of a molecule's constituent elements. However, this will generate less representative feature values as this would simply squash the atoms' features regardless of how their contribution might be with respect to the other atoms in the compound.

To construct a molecular representation that accurately reflects the relationships among the different constituent atoms and their contributions to both organic and inorganic atoms in the compound, we

developed a nested molecule-level graph representation. The nested graph is constructed from the SMILE representation of the organic molecules, with the same node features and edge features as the full material graph, as explained in Sections 3.3.3, 3.3.1. This nested graph is processed using a separate nested-GNN, as illustrated in Figure 1b on the right, that is built from a multi-head graph attention network (GAT) and trained in conjunction with the full graph GNN to encode representative molecular features. This nested GNN plays a role as a more sophisticated statistical merging method instead of manually merging to represent the organic molecules in the crystal structure so that more accurate and contextual molecular features can be generated.

### 3.3.3 CRYSTAL STRUCTURE INSPIRED EDGE-DESIGN

In a crystal graph, edges that describe the interaction between two neighboring atoms/molecules are usually defined by properties of chemical bonds, but these properties usually require complex DFT calculations to obtain. To facilitate the prediction and screening of novel HOIPs, we avoided using structural parameters. Instead, we derived interactive information between neighboring atoms/molecules directly from the chemical formula and designated this information as edge features.

To demonstrate this, in Figure 1b, we have depicted the ideal cubic structure of a mixed perovskite with the formula $MA_{0.7}FA_{0.2}Cs_{0.1}Pb_{0.4}Sn_{0.6}I_3$ as the crystal cell where constituent atoms/molecules of A, B, and X sites are probabilistically represented. For example, according to the provided formula, the probability of finding $MA$, $FA$, and $Cs$ in the vertices (A site) is 70%, 20%, and 10%, respectively. We use the term intra-group (local) ratio to describe these numbers, which should not be confused with the fractional ratio normalized across the entire compound. Thereafter, multiplying the intra-group ratios of two atoms/molecules from the same or different site groups naturally yields an interaction term reflecting the relative frequency (or average number of occurrences) at which the pair interacts in this cell representation. Hence, to reflect this relationship, we encode the product of the intra-group ratios of neighboring atoms/molecules as the edge weight or feature, which can be described as:

$$e_{(i,j)} = r_i \times r_j \tag{6}$$

where $r_i$ and $r_j$ are the intra-group ratios of two atoms/molecules represented by node $i$ and $j$, respectively. Accordingly, a dense undirected weighted graph is built for each perovskite, as depicted in Figure 1b on the right.

### 3.4 MODEL ARCHITECTURE

The overall architecture of our GNN is depicted in Figure 2. We utilized two distinct GNNs that operate hierarchically. The nested GNN (shown on the left in Figure 2) generates organic node features, while the $HP^3 - NS$ (shown on the right in Figure 2) is used to learn the overall graph representation and predicts the target material property's value. This is a unified model that can handle both inorganic and inorganic-organic hybrid materials.

The first part of the Hybrid GNN is the input to the convolutional layers. This input consists of two main components: the node features(either molecular or atomic) and edge features that describe the strength or weight of the connection between source node i and destination node j in the graph as described in Section 3.3.3. Edge weight features are firstly expanded using a Gaussian radial basis function (RBF):

$$e_{(i,j)}^{rbf} = exp(-\gamma(e_{(i,j)} - \mu)^2) \tag{7}$$

This expansion introduces non-linearity to improve the training process, resulting in a continuous uni-modal feature that can encode edge features with a desired resolution (Schütt et al., 2017). After that, node and edge features are embedded into higher-level features using separate Multi-Layer Perceptron (MLP). These MLPs are parametrized by $W_{edge}$, $W_{src}$, and $W_{dst}$, and are used to transform the input features into a new feature space where they can be more effectively processed. The transformed features are then concatenated together as,

$$z_{(i,j)} = W_{src}h_i||W_{dst}h_j||W_{edge}e_{(i,j)}^{rbf} \tag{8}$$

where $||$ denotes vector concatenation.

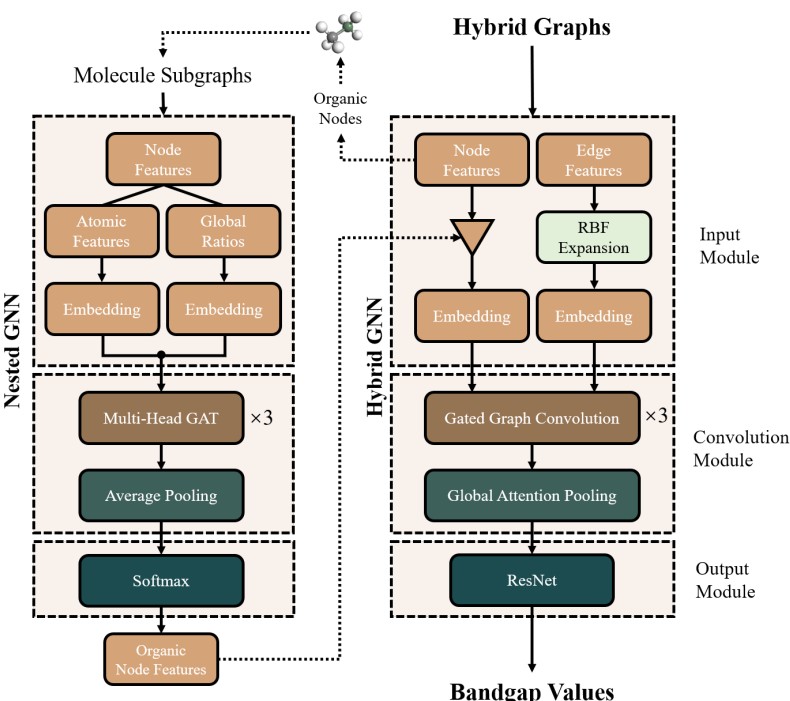

Figure 2: **The GNN architecture of our model.** While the layers on the left learn the molecular features, the layers on the right learn the overall graph's feature representation

To extract representations from the input graph, we used a variant of the GNN architecture that employs an attention mechanism (Veličković et al., 2018). It allows the model to selectively focus on the most relevant parts of the input graph for each node during the message-passing process so that the most informative features corresponding to the target material property will be highlighted in the model. Specifically, we modify the convolutional layers to incorporate an attention mechanism, which is implemented on the combined features $z_{(i,j)}$ to produce an attention coefficient $\hat{z}_{(i,j)}$:

$$\hat{z}_{(i,j)} = W_{gate} z_{(i,j)} \tag{9}$$

where $W_{gate}$ is an MLP. Once obtained, the attention coefficient is then normalized over all nodes $j$ using a SoftMax function:

$$\alpha_{(i,j)} = \frac{exp(\hat{z}_{(i,j)})}{\sum\limits_{k \in \mathcal{N}(i)} exp(\hat{z}_{(i,k)})} \tag{10}$$

where $\mathcal{N}(i)$ is the neighboring nodes of node $i$. Finally, each node acquires an attention score $\alpha(i,j)$ for each of its neighboring nodes, which determines the relative importance of the neighbor's information.

Then, each node's representation is updated as follows,

$$h_i^{t+1} = W_{dst} h_i^t + \sum\limits_{j \in N(i)} (w_1 \alpha_{(i,j)} + w_2 e_{(i,j)} + w_3 n_{(i,j)}) W_{src} h_j^t \tag{11}$$

where $w_1$, $w_2$, and $w_3$ are learnable scalar weights, and $h_i^t$ are the features of node $i$ at iteration $t$ with $h_i^0$ being the initial atomic feature of node $i$. To emphasize the contributions of stoichiometric features, both node weights $n_{(i,j)}$ and edge weights $e_{(i,j)}$ are weighted and added to the weighted attention score $\alpha_{(i,j)}$. The node weights $n_{(i,j)}$ were calculated by the presence probability of the target node as a neighbor of the source node, indicating the importance of a certain neighbor on the source node. Note that our gated convolutional layers update only node features, with edge features remaining unchanged. A Global attention pooling layer is employed as a readout layer on

the top of the convolutional layers to aggregate node representations into a high-dimensional graph-level representation. Following the readout layer, a residual network (He et al., 2016) with a skip connection is chosen as the head layer, following the work in Goodall & Lee (2020b) to predict the target property's value subsequently.

## 4 EXPERIMENT

We have selected the prediction of bandgap values in hybrid organic-inorganic perovskites as our target problem due to the availability of an open-source dataset and its practical relevance. This problem serves as a representative case for evaluating the effectiveness of our proposed technique in predicting the properties of other hybrid organic and inorganic materials. We have created a dataset of 933 HOIP samples from two data sources (Jacobsson et al., 2021; Li et al., 2019a), more details on this can be seen from Appendix A. The dataset is randomly split into training, validation, and test sets using 8:1:1 ratios.

To evaluate our HP$^3$-NS, we have selected four baseline models for comparison, namely Support Vector Regression (SVR), Crystal Graph Convolutional Neural Network (CGCNN), Roost (Goodall & Lee, 2020a), and CrabNet (Wang et al., 2021a). The first is selected because currently, the SOTA methods for predicting the properties of Hybrid organic and inorganic compounds are still classical machine learning methods, and CGCNN is selected as a representative of GNN models when the different issues are simplified using naive approaches, i.e., using statistically mixed features of constituent atoms to represent molecule features and using inter-ratio edge weights described in Section 3.3.3 as edge-weights in place of bond distance which requires the availability of crystal structure. The latter two are selected because they are state-of-the-art structure-agnostic GNN methods.

Different featurization techniques are applied, considering the specific input requirements of each model. For the classical tabular data-based models, we first obtained eight atomic properties (see Table 4) weighted by the stoichiometry of the constituent atoms in a perovskite material. These weighted properties were then concatenated statistically, taking into account their mean, standard deviation, maximum, minimum, range, and mean absolute deviation to represent the entire material. Furthermore, we combined the intra-group ratios of the 27 atoms/molecules shown in Figure 5b with the 48 statistically obtained features (eight atomic features times six statistics) obtained from the previous step to form the feature vectors (which we termed as tabular embeddings) for the three tabular data-based models. For CGCNN and HP$^3$-NS, the edge-featurization described in Section 3.3 is applied, and the node featurization, on the other hand, is detailed in Appendix B.

## 5 RESULTS AND DISCUSSION

### 5.1 HP$^3$-NS PERFORMANCE AND EFFECTIVENESS OF LEARNING MOLECULAR FEATURES

After training all the models, we computed the MAE for each model on the testing set. The results and corresponding input types are summarized in Table 1. Expectedly, the graph-based methods give better results and outperform the tabular data-based method SVR. Given the nature of the data, this superiority can be attributed to the inherent advantage of graph embeddings in representation learning. By modeling the problem as a graph representation learning, the model can learn effective representations crucial for predicting bandgaps. This is in contrast to relying on manually designed feature descriptors utilized by the traditional ML models, which lack direct relationships with the target property, bandgap, thereby hindering optimization and reducing overall performance.

The best (lowest MAE) result, Mean Absolute Error (MAE) of 0.0483 eV, is achieved when HP$^3$ is utilized alongside a nested graph neural network, HP$^3$-NS, that is jointly trained to represent molecular features from nested molecular graphs. This configuration yields a 7.5% improvement compared to the standalone HP$^3$-NS and a 10.7% improvement over CGCNN. Our method also outperforms similar state-of-the-art structure-agnostic methods, Roost and CrabNet, thereby substantiating the effectiveness of our proposed approach. The incorporation of sub-graphs to learn molecular features contributes to improved performance as it learns features in relation to their composition with other atoms from B and X sites and the target property's value—bandgap value in this instance. This is preferable to naively representing molecular features using a weighted mixture of atomic features, which can lead to ambiguously similar representations, especially since organic molecules are pri-

Table 1: Performance comparison of different models. The training parameters were set as default; all models were trained 300 epochs and compared.

| Models | Input Type | Datasets & Material Properties MAE | | | |
| | | Curated | DFT | | |
| | | Bandgap | Bandgap | Atomization Energy | Dielectric Constant |
| --- | --- | --- | --- | --- | --- |
| SVR | Tabular | 0.0608 | / | / | / |
| HP$^3$-NS | Nested Graph | 0.0483 | 0.336 | 0.0150 | 3.40 |
| Roost | Graph | 0.0622 | 0.343 | 0.0164 | 3.30 |
| CrabNET | Graph | 0.0515 | 0.373 | 0.0220 | 3.48 |
| CGCNN | Graph | 0.0541 | 0.352 | 0.0185 | 3.34 |
| CGCNN(with structure) | Graph | N/A | 0.141 | 0.0090 | 3.34 |

marily composed of carbon (C), hydrogen (H), oxygen (O), and nitrogen (N) atoms, and scarcely differ in ratios. In the curated dataset, the role of better organic molecule embedding is significant. Notably, Roost and CrabNet, lacking differentiation between intra-organic molecule interactions and crystalline interactions, exhibit inferior performance compared to our method. This underscores the significant contribution of our approach, which excels in capturing the nuanced interactions within these complex materials.

To facilitate a more robust comparison, we introduced another HOIP dataset (Kim et al., 2017) generated through DFT. This dataset offers a more extensive array of material properties along with calculated structural information, though it doesn't contain the mixing of perovskite crystalline sites. The original CGCNN, designed to accommodate structural data, was also added here as a benchmark for training and comparison, as shown in Table 1. In the DFT dataset, the role of organic molecules is more categorical since the dataset only contains the binary existence of organic molecules. So the advantage of our method is relatively limited. But it still outperforms both structure-agnostic methods. Besides, as for the CGCNN with structure models, it is an established fact that material properties are intricately tied to their structures, so it outperforms structure-agnostic methods in most properties. However, as elucidated in the introduction, structural data is often absent in conceptually designed new materials, especially when compositions are altered. The HP$^3$-NS approach facilitates large-scale and rapid screening of material properties, making it particularly advantageous in scenarios where structural data may be incomplete or unavailable, with reasonable compromise of accuracy.

## 5.2 ABLATION STUDY

To validate the effectiveness of the components in the proposed method, we did an ablation study by removing certain components of the graph representation design, namely, the nested subgraph, the node weights, and the inta-group ratios as edge design. The results are presented in Table 2. From the ablation study, we can identify the major effect of intra-group ratios and nested subgraphs. The pivotal role of intra-group ratios lies in their integral contribution to the edge design within our structure-agnostic graph representation. The nested graph component corresponds to the heterogeneous embedding of organic molecules within crystalline structures. The differentiation in handling organic molecules and atoms theoretically enables a more accurate embedding of their roles in crystal structures. Besides, node weights are the ratios of interactions among nodes at the crystal level and serve to emphasize the importance of more prevalent atoms or molecules when updating a node's features. While the introduction of similar information about statistical mixing has been addressed by intra-group ratios, the contribution of node weights is relatively limited. Nonetheless, it still enriches the model by incorporating additional knowledge about the constrained ABX$_3$ structure and the relative importance among A, B, and X sites within the crystal.

Table 2: Performance comparison between $HP^3 - NS$ and the models without certain components

| Models | Bandgap MAE |
|---|---|
| HP3-NS | 0.0483 |
| Without nested graph | 0.0522 |
| Without intra-group ratios | 0.0534 |
| Without node weights | 0.0499 |

### 5.3 LEARNED GRAPH EMBEDDINGS

To demonstrate the effectiveness of the proposed representation learning, a more detailed interpretation of the learned graph embeddings $h_i^t$ is crucial. Hence, We applied dimension reduction on the high-dimensional embedding space to visualize better what is learned using our graph representation approach in comparison to others (Figure 3. Specifically, we utilized t-distributed stochastic neighbor embedding (t-SNE), effectively reducing high-dimensional features into lower dimensions while preserving the features' local and global structure. We observed that compared to the manually designed tabular embeddings mentioned above, the HP³-NS graph embeddings exhibit a more ordered spatial relationship with respect to their corresponding bandgaps. The distributions of the embeddings are also more compactly classified, indicating a more adequate and efficient encoding of information than the tabular embeddings and the CGCNN model.

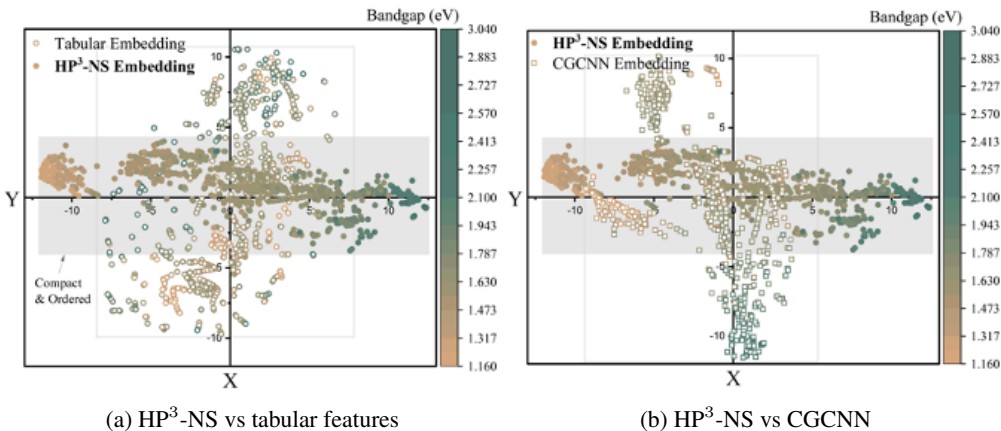

(a) HP³-NS vs tabular features      (b) HP³-NS vs CGCNN

Figure 3: Comparison of tabular embedding space and CGCNN embedding with our HP³-NS embedding space as visualized using tsne

Furthermore, as an MAE of 0.0483 ev is a reasonable error deviation for HOIP material's bandgap prediction value, we fabricated a series of perovskite films that were not present in the existing dataset and measured their bandgaps to validate the model's predictions to further verify our model's applicability in real-world scenarios. The synthesis details of perovskite films can be found in Appendix D.2. Particularly, we synthesized 35 new perovskites with the $MA_xFA_yCs_{1-x-y}Sn_mPb_{1-m}(Br_nI_{1-n})_3$ configuration and examined their UV-vis absorption spectra to determine the bandgap values. The experiment sample selection procedure followed can be seen in Appendix D.1

Figure 4a displays a scatter plot of the predicted bandgap values against the measured values of the newly synthesized perovskite materials. Note that experimental data obtained from the test set is also included in the plot. As shown in Figure 4a, most scatter points cluster around the diagonal line, indicating a high level of agreement between the predicted and measured results. Furthermore, the MAEs for all the experimental and newly synthesized data were as low as 0.042 eV and 0.043 eV, respectively. These results demonstrate the precise prediction of bandgaps for $MA_xFA_yCs_{1-x-y}Sn_mPb_{1-m}(Br_nI_{1-n})_3$ materials by HP³-NS. Typically for deep learning

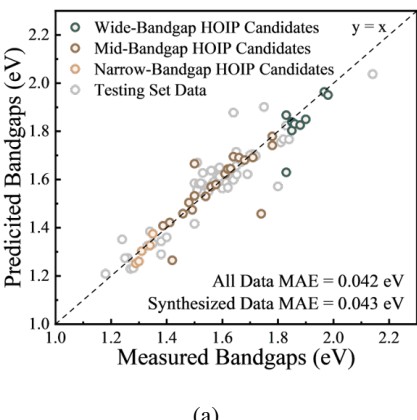 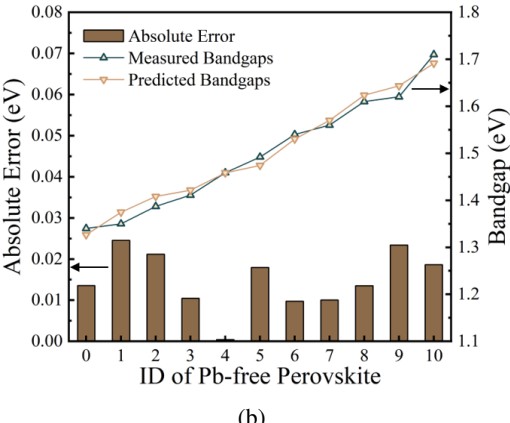

(a)             (b)

Figure 4: Experimental evaluation of the performance of HP$^3$-NS. (a) depicts a comparison between the predicted and measured bandgaps of the 108 mixed HOIP material. It presents two types of mean absolute errors: one calculated using all the data and the other using the synthesized data. (b) depicts the absolute errors between the predicted and measured bandgaps of the 11 lead-free perovskites

models, our curated dataset, which only contains 900 entries, is considered a small data set. However, the minor deviations observed between predictions and measurements suggest that this dataset is sufficient for our model to grasp the underlying relationship between the HOIP features and their corresponding bandgap values.

We also conducted further evaluations to assess the performance of our model on a subset of 11 lead-free perovskites out of the 35 perovskites synthesized. It is worth noting that the presence of lead-free materials in our collected dataset is limited to only 10%, which presents a challenge for accurate property prediction. Nonetheless, all 11 lead-free compounds exhibit absolute errors below 0.025 eV, Figure 4b, highlighting the potential of our model to estimate lead-free perovskites' bandgaps accurately. Further analysis and application of the model can be found in Appendix D.3

## 6 CONCLUSION

This study presents a unified model, HP$^3$-NS, that can accurately predict the properties of inorganic and hybrid inorganic-organic materials using a graph representation methodology. We learned organic molecular features by using nested graph representation learning, which resulted in improved prediction accuracy as validated both using empirical results and actual experiments. Our HP$^3$-NS has outperformed all the commonly used traditional methods and improved over a representative GNN model that uses naive molecule representation by over 10%. Furthermore, we circumvent the requirement for crystal structure data by incorporating structure representation through edge encoding. This modification enables our model to be employed for large-scale material exploration and screening, broadening its applicability in material science.

In perovskite application, we utilized HP$^3$-NS to discover environmentally friendly perovskite materials for solar cell design. This method presents a promising avenue for future research as it can save computing or experimental resources to help researchers navigate complex material properties within vast and complex composition space like a map.

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

## APPENDIX A  DATASET PREPARATION

The HOIP dataset is curated from an open-source, high-quality experimental bandgap database collected as part of a Perovskite Database Project (Jacobsson et al., 2021), which has over 42,000 entries from peer-reviewed literature. Moreover, we have also added some entries from Li et al. (2019a) that have over 300 entries collected from over 2000 literature works.

After acquiring the raw data from these two sources, we preprocess it by removing duplicates, entries with missing target values, and samples containing rare atoms/molecules or incorrect chemical formulas. As a result, we are left with a dataset comprising 933 unique and clean samples. Figure 5a illustrates the distribution of perovskite bandgaps in the processed dataset, which spans from 1.16 to 3.04 eV, with approximately two-thirds of the data falling within the range of 1.50-1.90 eV. Figure 5b shows the 27 distinct types of atoms/molecules that occupy the A, B, or X sites within the perovskite crystal structure, along with their respective frequencies, as observed in our dataset. These figures reveal that the majority of perovskite materials in our curated dataset contain lead. This category accounts for almost 90%

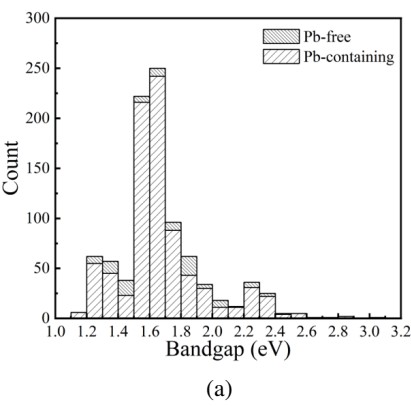
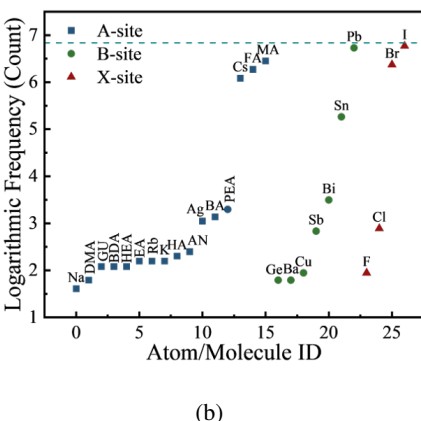

(a)                       (b)

Figure 5: Distribution of the experimental HOIP dataset. (a) Histogram of bandgap distribution in the dataset. Each bar is divided into two segments: the lower one represents lead-containing perovskites, while the upper one represents lead-free perovskites. (b) Frequency of atoms or molecules from different sites in the dataset. If a perovskite material contains a particular type of atom or molecule, the frequency of that atom or molecule would increase by one. Then, we take the logarithm to compress the y-axis scale to visualize the relative differences better between the different frequencies. The dashed line in (b) indicates the total amount of our data in the dataset.

## APPENDIX B  HYPER-PARAMETER SELECTION

We have applied hyper-parameter tuning on all the models to find the best combination of hyper-parameters that results in the lowest MAE for each of the models. We utilized search and early-stop algorithms from Ray Tune (Liaw et al., 2018) to determine the optimal hyperparameter configuration of our approach $HP^3$-NS model. The best model was trained for 300 epochs with a batch size of 64, using MAE as the loss function and AdamW with a learning rate of $2 \times 10^{-3}$ and a weight decay of $10^{-5}$ as the optimizer. The hyper-parameters selection space and the algorithms used to search the optimal configuration on each of the different ML models are shown in Table 3.

For models that rely on graph-structured data, such as CGCNN and our model, we converted all atomic features into categorical features, following the approach outlined in CGCNN (Xie & Grossman, 2018b). Specifically, we applied a one-hot encoding scheme for features with finite values of less than 20 levels to represent each level as a separate category. For features with 20 or more levels, we divided the range into ten bins and then assigned each bin to a category. While CGCNN utilizes uniform binning, which divides the feature range into evenly spaced intervals, we found that this approach can generate empty categories and a skewed distribution of categories, which could potentially mislead the ML model and adversely impact its performance, as it may result in atoms

| ML algorithm | Hyper-parameter search space | Search algorithm |
|:---:|:---:|:---:|
| LR | - | - |
| SVM | $C \in [0.01, 100]$,
degree $\in \{3, 4, \ldots, 6\}$
gamma $\in [0.01, 10]$
kernel $\in \{poly, linear, rbf, sigmoid\}$
epsilon $\in [0.01, 10]$ | RandomizedSearchCv-scikitlearn |
| LGBM | max depth $\in \{5, 6, \ldots, 8\}$
min child weight $\in [0.001, 100]$
learning rate $\in [0.0001, 0.1]$ | TuneSearchCV-tunesklearn |
| CGCNN, $HP^3$-NS | batch size $\in \{1, 32, 64, 128\}$
conv layers $\in \{2, 3, \ldots, 5\}$
linear layer dim $\in \{64, 128, 256\}$
dropout $\in [0.05, 0.15]$
activation function $\in \{relu, silu, mish, sigmoid\}$
lr $\in [1e-4, 2e-3]$
weight decay $\in [1e-5, 1e-4]$ | HyperOptSearch-raytune |

Table 3: Hyper-parameter search space and algorithms used for the different algorithms

with unique properties being grouped together, giving rise to the false assumption that they share similar or identical properties. To address this issue, we implemented non-uniform binning, which allowed us to create a balanced distribution of categories by leveraging quantiles, as demonstrated in Figure 6.

## APPENDIX C    MAGPIE ATOM PROPERTIES LIST

Table 4 in the appendix contains all the atomic properties that are extracted to create an atomic feature of a particular atom in Materials Agnostic Platform for Informatics and Exploration (Magpie). Although Magpie's original paper considers more than 22 different atomic properties, only a few are included here after using domain knowledge to remove unrelated properties and those with missing values.

## APPENDIX D    EXPERIMENTAL VERIFICATION

### D.1    EXPERIMENTAL DESIGN AND SAMPLE SELECTION

We selected $MAI$, $FAI$, and $CsI$ to adjust the ratio of A-site cations. Similarly, $PbI_2$, $SnI_2$, $PbBr_2$, and $SnBr_2$ were used to adjust the ratio of B-site and X-site cations. To ensure that the selected candidate samples have good structural stability, we used tolerance factor (Tf) and octahedron factor (Of) as criteria (Tf between 0.8 and 1.2, Of between 0.4 and 0.7) for further screening. Based on the model-predicted bandgaps, we categorized the potential candidates into three groups, representing narrow, medium, and wide bandgap levels. Finally, we selected 35 samples with diverse bandgap levels for synthesis. Considering the dataset's limited representation of lead-free perovskites, we intentionally included 11 lead-free samples. These lead-free perovskites pose a challenge for prediction and serve as valuable benchmarks to evaluate the performance of ML predictors.

### D.2    EXPERIMENTAL PROCEDURES

Perovskite Precursor Solution Preparation: 1.1 M precursor solutions for all perovskite films are prepared by dissolving all or part of MAI (Macklin, 99.99%), FAI (J&K Scientific, 99.99%), CsI

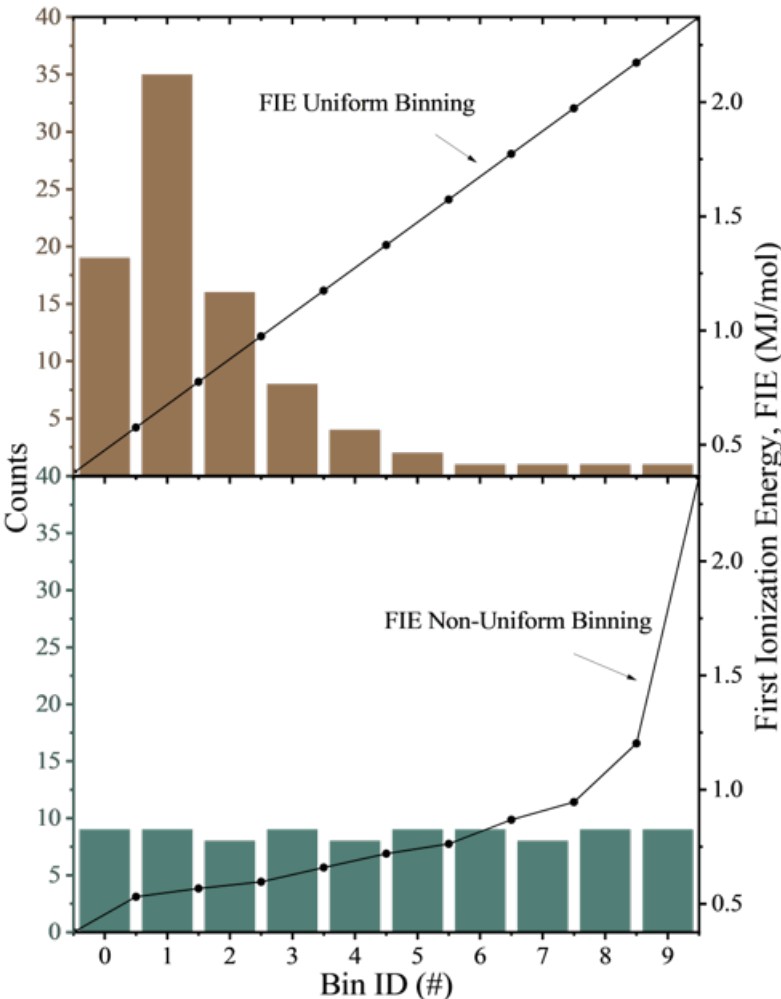

Figure 6: Comparison of uniform and non-uniform binning for categorical representation of elemental features. We use the first ionization energy as an example feature to illustrate the influence of different binning approaches.

Table 4: Comprehensive overview of node features used in our study. We adopted eight atomic features from Magpie and one stoichiometric feature as the initial input for each node in the perovskite graphs. The atomic features were encoded using a one-hot encoding technique similar to the one used in CGCNN (Xie & Grossman, 2018b). Further information on the one-hot encoding process can be found in section c of this paper.

| Source | Feature | Description |
|---|---|---|
| Magpie | tomic number | The number of protons in an atom |
| | Atomic radius | The size of an atom |
| | Atomic weight | The average mass of an atom |
| | Electronegativity | The ability of an atom to attract electrons |
| | First ionization energy | The energy required to remove an electron from an atom |
| | Group/Family | The vertical column of the periodic table |
| | Molar volume | The volume occupied by one mole of a substance |
| | Period | The horizontal row of the periodic table |
| Stoichiometry | Fractional ratio | The proportion of atoms of one element in a compound or mixture |

(J&K Scientific, 99.99%), PbI2 (Macklin, 99.999%), PbBr2 (Macklin, 99.999%), SnI2 (J&K Scientific, 99.999%) and SnBr2 (J&K Scientific, 99.999%) in a mixed solvent of DMF (Sigma-Aldrich, anhydrous) and DMSO (Sigma-Aldrich, anhydrous) with a volume ratio of 4:1 according to the stoichiometric ratio. The precursor solution is heated to 60°C until completely dissolved and then filtered through a nylon 66 filter (pore size 0.22 $\mu m$) before being used for film preparation. All the above operations are carried out in a nitrogen-filled glovebox.

Perovskite Films Fabrication: $12mm \times 12mm$ quartz substrates are ultrasonically cleaned in deionized water, ethanol, and acetone for 30 min, then blown dry with nitrogen, and then treated under a UV-Ozone (SETCAS LLC, SC-UV-I) for 30 min to remove organic residues. Deposition and annealing of perovskite films are carried out in a nitrogen-filled glovebox. 40 $\mu L$ precursor solution is spin-coated in a two-step procedure at 1500 and 5000 rpm for 15 s and 40 s, respectively, and 200 $\mu L$ of chlorobenzene (Macklin, anhydrous) is dropped on the spinning substrate 10 s before the end of the procedure. The films are then annealed on a hot plate at 150°C for 15 min.

Perovskite Films Characterization: Absorption spectrums of the perovskite films on quartz substrates are measured by a UV-VIS-NIR spectrophotometer (JASCO V-770). Tauc-plots are generated from the absorption spectra to obtain the bandgaps of perovskite material with different stoichiometric ratios.

Model Assessment: Experimental and predicted bandgap values are compared using mean absolute error (MAE):

$$MAE = \frac{\sum_{i=1}^{N} |y_i - \hat{y}_i|}{N} \tag{12}$$

where N denotes the number of samples, $y_i$ denotes experimentally measured bandgap values, and $\hat{y}_i$ denotes predicted bandgap values.

### D.3 MATERIAL ANALYSIS AND SCREENING WITH HP³-NS

Based on the results mentioned above, it can be observed that HP³-NS can accurately map the composition of HOIPs to their bandgap values without the need for structure-related features. While GNN models relying on structural information may offer better predictive accuracy, they face challenges when predicting a large number of new materials. This is because obtaining the crystal structure information of these materials is computationally expensive to prepare for all possible materials

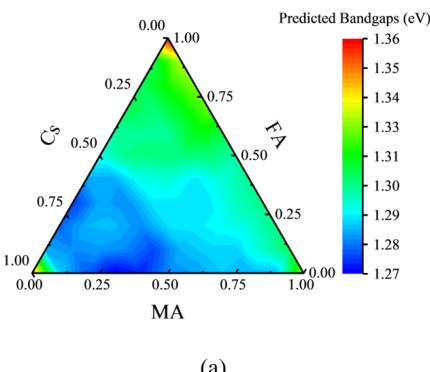
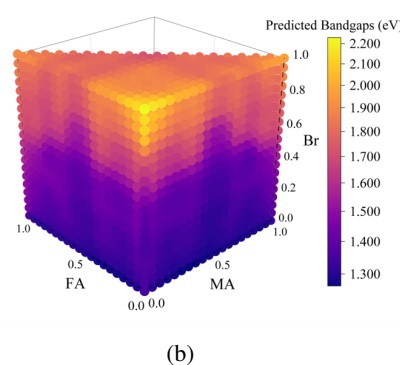

(a)                                        (b)

Figure 7: large scale analysis result (c) depicts a ternary contour plot illustrating the impact of variations in the ratio of the three A-site cations on the bandgap of $MA_xFA_yCs_{1-x-y}SnI_3$. (d) depicts a 3D graph illustrating the bandgap predictions of HP³-NS across the compositional space of lead-free perovskites $MA_xFA_yCs_{1-x-y}Sn(Br_nI_{1-n})_3$.

at inference time. Hence, there is always a need for a structure-agnostic model that can be used for large-scale screening and analysis. Given our HP³-NS doesn't need crystal structure information, it has great potential to work as a map to help researchers navigate material properties in vast design spaces after only training a small amount of prior labeled data.

With its high level of accuracy and structure-agnostic design, HP³-NS is a convenient tool for material analysis and screening. An example is demonstrated by using our model to analyze the impact of A-site cation concentrations on the bandgaps of $MA_xFA_yCs_{1-x-y}SnI_3$, as depicted in Figure 7a. The ternary contour plot highlights that an increase in Cs concentration results in a decrease in bandgap values, while an increase in MA or FA produces the opposite effect. By leveraging this tool, we can predict the minimum and maximum bandgaps for this perovskite type, which are 1.27 eV and 1.36 eV, respectively. The minimum bandgap is achieved when the A-site cations consist of MA and Cs in a ratio close to 3:7, whereas the maximum is observed when only FA is present. This analysis can be extended to examine different intricate relationships, such as analyzing how different constituent atoms or molecules affect the properties of other perovskite configurations.

Additionally, HP³-NS excels in efficiently screening a vast number of candidate materials, which is crucial for accelerating the material discovery process. For example, our model enables the prediction of bandgaps for Sn-based perovskites $MA_xFA_yCs_{1-x-y}Sn(Br_nI_{1-n})_3$, encompassing all possible lead-free materials derived from the seven chemicals applied in this study for experimental synthesis. Since the ratios of two of the three A-site cations and one of the two X-site halide anions are the only adjustable variables, the predicted results for all possible combinations can be visualized through the 3D graph depicted in Figure 7b. The graph reveals a broad bandgap range of 1.3 eV to 2.2 eV for $MA_xFA_yCs_{1-x-y}Sn(Br_nI_{1-n})_3$, meeting the bandgap requirements for perovskite layers in PSCs. For example, in a typical two-terminal tandem solar cell, the bottom cell usually requires a bandgap spanning 1.2 eV to 1.3 eV, while the top cell necessitates a bandgap ranging from 1.7 eV to 1.9 eV. The predictions from HP³-NS indicate the existence of numerous potential lead-free candidates falling within these specified ranges. Based on this pool of candidates, further investigations can be conducted, such as developing environment-friendly PSCs with promising lifetimes and efficiencies.

