# OpenReview forum: "HP$^3$-NS: Hybrid Perovskite Property Prediction Using Nested Subgraph"
_ICLR.cc/2024/Conference — ICLR 2024 Conference Withdrawn Submission_

### Official Review · Reviewer_Std6 · 2023-10-25

**Soundness:** 3 good
**Presentation:** 3 good
**Contribution:** 3 good
**Rating:** 5
**Confidence:** 4

**Summary:**

This paper proposes a hybrid graph neural network (GNN) architecture to predict the properties of hybrid materials comprising both organic molecules and inorganic atoms. The proposed method shows effectiveness in predicting bandgap value for organic-inorganic perovskites.

**Strengths:**

1.	This work aims to leverage GNN to predict the properties of hybrid organic and inorganic materials. This is a novel task.

2.	This work proposes using a hybrid GNN to obtain high-quality graph-level embeddings for hybrid materials. The proposed hybrid GNN contains a nested GNN to specifically extract molecular representation for the organic molecule contained in the material. This model design is interesting.

3.	A new dataset of organic-inorganic perovskites is proposed, which can facilitate future ML works studying this task.

**Weaknesses:**

1.	Does the curated dataset provide atom coordinates?

2.	Why do authors follow classic ML methods to do structure-agnostic prediction? Currently, 3D structures are commonly considered by GNN methods developed to predict material properties. I don’t think “aiming to apply our model for analyzing large scale materials and discovering novel materials” and “using 3D structure” are contradictory. If the dataset contains material structures, and the proposed method can outperform SOTA methods like CGCNN (using structure). Then, it’s an exciting result.

3.	An ablation study is needed to confirm the effectiveness of the proposed edge design.

4.	Does the curated dataset provide more property targets? It’s encouraged to do experiments with more properties.

5.	In the last equation in Sec 3.4, what are node weights $n(i, j)$? Its definition is missing.

6.	Minor: The serial numbers for equations are missing from Page 5.

7.	Minor: In Figure 1(b), the $e(i, j)$ between Cs and Sn is wrong. The ratios of H and N in $CH(NH_2)_2$ are switched.

8.	Typo: In Sec 3.2, “Each of the A, B, and C can be …” Here, C should be X?

**Questions:**

See in weakness.

---

> ### Author Response · Authors · 2023-11-22
> **Thanks for the advices**
>
> We appreciate the time and effort you have dedicated to evaluating our manuscript. We are grateful for your insightful comments, which we believe have contributed significantly to the refinement of our work. Below, we provide responses to your comments and address the concerns.
>
> > Reviewer’s Question: Does the curated dataset provide atom coordinates?
>
> Thanks for the question. We would like to clarify that our dataset does not include atom coordinates. The decision to curate a dataset with pure experimentally collected perovskite data was made to avoid bias introduced by different DFT-calculated results. Experimental datasets typically lack atom coordinates due to the difficulty of experimentally characterizing them. We will ensure this is explicitly mentioned in the revised manuscript.
>
> > Reviewer’s Question: Why do authors follow classic ML methods to do structure-agnostic prediction? Currently, 3D structures are commonly considered by GNN methods developed to predict material properties. I don’t think “aiming to apply our model for analyzing large scale materials and discovering novel materials” and “using 3D structure” are contradictory. If the dataset contains material structures, and the proposed method can outperform SOTA methods like CGCNN (using structure). Then, it’s an exciting result.
>
> We acknowledge the importance of 3D structures and emphasize that our intention is not to dismiss their relevance. However, 3D structure data, i.e., atom coordinates, are generally not available for novel materials designed conceptually in large-scale materials screening. Material scientists have two approaches to obtain structure data for a novel material. The first is to carefully synthesize and characterize the material experimentally with sophisticated equipment. However, this approach is impractical when screening materials from a pool of millions of candidates. The second approach is to calculate atom positions with DFT by determining the most stable positions, which is how most 3D structure data are obtained in materials datasets that are used extensively by GNN methods.
>
> Nevertheless, these calculations are computationally expensive, especially for large crystal super-lattices. For example, in our case, when the perovskite ABX3 is doped with a low concentration of organic cations A’, A’_0.02A_0.98BX3, to represent these low concentrations in atoms with coordinates, thousands of atoms in the super-lattice may be needed, requiring a supercomputer cluster to solve. Hence, we assert that using a 3D structure in large-scale material screening is not feasible, particularly in our case.
>
> While we agree that it would be exciting if the proposed structure-agnostic method could outperform state-of-the-art methods that use structure data, it is essential to recognize that structure data are indeed crucial in understanding materials' properties and adhering to physical laws (The comparison is included in the general reply for baselines). The contribution of our method lies in achieving accurate predictions without the explicit use of detailed structural information, thereby facilitating the model's application in large-scale novel materials screening and discoveries.
>
> > Reviewer’s Question: An ablation study is needed to confirm the effectiveness of the proposed edge design.
>
> We appreciate the suggestion for an ablation study to confirm the effectiveness of our proposed edge design. In the revised manuscript, we will include an ablation study to provide a more in-depth analysis of the model components and their impact on performance. Most reviewers share the concern about the lack of an ablation study, so we posted the extended experiment results in a general reply.
>
> > Reviewer’s Question: Does the curated dataset provide more property targets? It’s encouraged to do experiments with more properties.
>
> We agree with this suggestion and will conduct additional experiments on our dataset to explore predictions for multiple property targets. Since most reviewers share this concern, we will post a general reply with the results. We hope the extended experiments will address your concerns.
>
> > In the last equation in Sec 3.4, what are node weights $n_(i,j)$? Its definition is missing.
>
> Thanks for the question. We are sorry for the ambiguity. The node weights $n_(i,j)$ are calculated by the fractions of the node's presence in the crystal as the source node $n_i$ ‘s neighbor, which indicates how often the node will interact with the source node.  This metric is rescaled and added to the overall weight for a specific neighbor to evaluate its effect on the source node. We will include the definition in the revised manuscript.

---

> > ### Author Response · Authors · 2023-11-22
> > **Continued**
> >
> > > Minor: The serial numbers for equations are missing from Page 5.
> >
> > > Minor: In Figure 1(b), the  between Cs and Sn is wrong. The ratios of H and N in  are switched.
> >
> > > Typo: In Sec 3.2, “Each of the A, B, and C can be …” Here, C should be X?
> >
> > Thanks for kindly pointing out the typos and formatting errors in our manuscripts. We have carefully revised the manuscript to address all identified typos and to enhance overall clarity. We are confident that these revisions significantly improve the readability and overall quality of the manuscript.

---

### Official Review · Reviewer_nTVd · 2023-10-27

**Soundness:** 2 fair
**Presentation:** 2 fair
**Contribution:** 2 fair
**Rating:** 5
**Confidence:** 3

**Summary:**

This paper proposed a method to construct graph connections and edge features for HOIPs, which can be represented as $ABX_3$, without structural information, to capture the relationships between atoms. The experiments show that the proposed graph construction is better than direct merging all features.

**Strengths:**

1. The authors claim to be the first work to utilize GNN and to learn separate representations for organic molecules.
2. They collected a small dataset with ~900 HOIP samples.

**Weaknesses:**

However, I hold concerns for the significance and generality of the proposed method.

1. The major contribution is a graph construction method without structural information, basically how to determine the edge connections and how to determine edge features, for a constrained  HOIPs system with format $ABX_3$. The solution is somehow straight forward and may not be able to extend to general cases.

2. Although there are limited approaches for HOIP systems, but constructing heterogeneous graphs to capture interactions between different components of chemical systems is not new, and this proposed method is far from the first.

**Questions:**

Is the constructed graph a fully connected one? The graph construction details are not very informative, and the figures in the paper is not very clear.

**I have read the authors’ rebuttal, and the concerns remain.**

---

> ### Author Response · Authors · 2023-11-22
> **Thanks for the advices**
>
> > Graph Construction Details:
>
> Thanks for your question. Yes, the constructed graph is fully connected, with edges representing the interactive information in the crystal calculated from the crystal symmetry and the chemical formula. We apologize for any confusion regarding the clarity of the graph construction details. In the revised manuscript, we will provide a more comprehensive explanation of the graph construction methodology, including a clearer presentation of the figures.
>
> > Regarding the Concerns About Significance and Generality:
>
> We appreciate the attention to the significance and generality of our proposed method. We acknowledge that our approach focuses on a specific HOIPs system with the format ABX3. The reason we focused on this system, on the one hand, is that it has shown promising potential as a semiconductor material with applications in various fields. On the other hand, we identified that a more common issue of the missing structural data for conceptually designed novel materials has impeded the application of GNN models in real-case material screening. By designing the $HP^3-NS$ graph representations, we facilitated this process and validated the usage with real material synthesis and measurements.
>
> While we acknowledge that its extension to general cases may require further investigation, we believe the proposed method to construct nested graphs for hybrid crystalline materials systems and use the stoichiometry formula to construct the edges and models can be adapted to other hybrid or pure inorganic crystalline materials of interest. The $ABX_3$ constraint was actually some prior knowledge about the symmetry of the lattice that can be generalized to other lattice constraints.
>
> > Regarding the Concerns About Originality:
>
> We appreciate the reviewer's comments on the originality of our work. While we acknowledge that the idea of utilizing heterogeneous graphs has been applied in some works to address specific domain problems, we want to emphasize our unique contribution to dealing with hybrid systems in crystalline materials. Specifically, our edge design does not rely on structural data, addressing a significant challenge when handling complex systems with organic molecules where detailed information is often lacking.
> In our approach, we have to constrain the model with certain symmetry knowledge of the lattice due to the absence of structural data. We initiated the graph construction by introducing a new edge design and nested graph inspired by the physical interactions in crystalline materials. The effectiveness of these proposed components has been demonstrated in the ablation study and benchmarking.
>
> To the best of our knowledge, there is no similar structure-agnostic Graph Neural Network (GNN) work focused on modeling hybrid organic-inorganic systems. Moving forward, we plan to conduct more generalized and systematic work on refining and formalizing our model. We appreciate the valuable feedback provided and hope that our perspective on addressing the unique challenges in the representation of hybrid systems in crystalline materials is considered.

---

> ### Author Response · Authors · 2023-11-23
> **Regarding the Concerns About Generality**
>
> This method works for more kinds of hybrid organic-inorganic crystalline materials in addition to pervoskite materials, where existing solutions are limited.
> 1. Metal-Organic Frameworks
> 2. Hybrid Zeolites
> 3. Organic-inorganic halide semiconductors
> 4. Layered hybrid materials
> 5. Hybrid Ferroelectrics
> and so on.

---

### Official Review · Reviewer_su6c · 2023-10-31

**Soundness:** 2 fair
**Presentation:** 2 fair
**Contribution:** 2 fair
**Rating:** 5
**Confidence:** 3

**Summary:**

The authors propose a nested graph representation for perovskite property prediction.

**Strengths:**

1. The investigated problem is important in the material science domain.
2. The figures in the experiment section are illustrative.

**Weaknesses:**

1. More baselines should be used and tested on more datasets.
2. More technical details should be included.
3. More theoretical contributions should be made.

**Questions:**

1. Are there any more baselines that should be compared?
2. What are the theoretical contributions of this work?

---

> ### Author Response · Authors · 2023-11-22
> **Thanks for the advices**
>
> We appreciate the time and effort invested in reviewing our manuscript. We are grateful for the constructive feedback and would like to address the concerns raised in the review.
>
> > Reviewer Comment: More baselines should be used and tested on more datasets.
>
> We acknowledge the importance of comprehensive evaluation and agree that including additional baselines and testing on diverse datasets can enhance the robustness of our proposed method. In response, we have expanded the set of baseline models in the revised manuscript. We have mainly chosen structure-agnostic methods as baselines, aligning with the nature of our problem. Furthermore, we have extended our experimentation to cover a more extensive range of datasets and material properties. We will provide a detailed discussion of baseline selection and result in a general reply to reviewers, as several raised similar concerns. We believe these additions will substantially address the raised concerns and provide a more thorough assessment of our methodology.
>
> > Reviewer Comment: More technical details should be included.
>
> Thank you for your request for more technical details. Dealing with hybrid organic-inorganic material systems in the absence of structural data requires a meticulous approach. We carefully formulated the problem constraints, constructed material graphs, designed edges inspired by crystal lattice interactions, specified node details, selected and binned node features, and built subgraphs. Unlike models with structural data inputs, our approach indirectly incorporates lattice knowledge by differentiating intra-group and inter-group ratios (node weights). Our model design employed different Graph Neural Network (GNN) structures for the crystal graph and the nested subgraph, tailored to their respective purposes. The specific technical details are elucidated in dedicated sections within the Methods part of the revised manuscript.
>
> We apologize for the ambiguity in how we formatted these technical details. In the revised manuscript, we will provide additional technical information, including a more in-depth description of the nested graph representation, training procedures, and any other relevant technical aspects in the appendix sections. Including the detailed parameters we used to train the models. This will ensure clarity and help readers better understand the methodology.
>
> > Reviewer Comment: More theoretical contributions should be made.
>
> We recognize the importance of clearly articulating the theoretical contributions of our work. As an application work, we would like to emphasize our contribution to tackling the specific challenges posed by hybrid organic-inorganic crystalline materials in practical applications, where large-scale material candidates with complex doping levels are present but structural data are unavailable. As a generalization, we formulated the hybrid organic-inorganic crystalline material systems problem as graph property predictions given no structural information, which poses the problem of proper edge design and graph construction. The proposed edge design and nested graph construction were inspired by the actual interactions in crystalline lattices in materials, as our proposed contribution of graph design in hybrid materials representations. The $ABX_3$ constraint was actually another source of information injected inherently as pre-knowledge about the symmetry of the lattice.

---

### Official Review · Reviewer_C55J · 2023-10-31

**Soundness:** 3 good
**Presentation:** 2 fair
**Contribution:** 3 good
**Rating:** 5
**Confidence:** 4

**Summary:**

This paper proposes a novel graph representation for hybrid inorganic-organic materials targeted for perovskite bandgap prediction. Specifically, the proposed method considers the representation of molecules with nested graphs. And it attains the edge features from the chemical formula directly, eliminating the need for computationally intensive DFT calculations. Notably, the authors synthesized and tested 35 new perovskites, demonstrating a strong alignment between experiment results and model predictions.

**Strengths:**

1. The problem of hybrid organic-inorganic (HOI) crystalline materials is interesting, and the motivation is convincing.

2. The method is reasonable, and extracting the edge features directly from the chemical formula is a desirable property, which is necessary for material discovery.

3. The synthesis experiment really helps validate the proposed method and make it more convincing.

**Weaknesses:**

1. The experiments are kind of inadequate, as the GNN baselines are too old and few (only 1 GNN...), and there are no ablation studies.

2. Writing typos. The paper needs more careful proofreading.

**Questions:**

Please answer the first question in the weakness part.

---

> ### Author Response · Authors · 2023-11-22
> **Thanks for the advices**
>
> We appreciate the thorough review of our manuscript and the valuable feedback you provided. We have carefully considered each comment and made corresponding revisions to address the concerns raised. Below, we provide a detailed response to each point:
>
> > Reviewer's Comments: The experiments are kind of inadequate, as the GNN baselines are too old and few (only 1 GNN...), and there are no ablation studies.
>
> We appreciate your consideration and shared your concern regarding the adequacy of our experimental evaluation in the initial manuscript. Our primary objective in the initial submission was to showcase the application of $HP^3-NS$ in actual material property prediction and screenings. We realized the importance of providing a more comprehensive comparison with multiple baselines and conducting ablation studies to establish the superiority of our representation design.
>
> Taking your valuable advice into account, we have taken steps to address these concerns in the revised manuscript. The experimental section has been expanded to include additional state-of-the-art graph neural network (GNN) baselines that are particularly relevant to the hybrid perovskite prediction task. Furthermore, we have incorporated ablation studies to thoroughly analyze the impact of individual components of our proposed method on predictive performance.
>
> **Considering similar questions posed by other reviewers, we plan to present the results of these extended experiments in a general reply, ensuring that the insights gained are accessible to all reviewers**. We sincerely hope that these additional experiments will sufficiently address your concerns regarding the perceived limitations in our initial experimental setup. We are grateful for your guidance and look forward to your feedback on the revised manuscript.
>
> > Reviewer's Comments: Writing typos. The paper needs more careful proofreading.
>
> We appreciate the reviewer's feedback on the need for improved proofreading. We have carefully revised the manuscript to address all identified writing typos and to enhance overall clarity. We have also engaged professional proofreading services to ensure a higher standard of language and presentation. We are confident that these revisions significantly improve the readability and overall quality of the manuscript.

---

### Author Response · Authors · 2023-11-22
**General Reply: Suggestion on the Inclusion of Additional Baselines and Datasets #1**

We deeply appreciate the time and effort invested by each reviewer in thoroughly critiquing our manuscript. Your insightful feedback has proven invaluable, and we are fully committed to enhancing the rigor and quality of our work based on your constructive commentary. The prevalent concerns pertaining to the paucity of supplementary baselines, datasets, and ablation studies have been duly registered and comprehensively redressed as elucidated below:

## __Suggestion on the Inclusion of Additional Baselines and Datasets:__
We appreciate the suggestion to include more diverse state-of-the-art baselines to facilitate a more holistic and unbiased evaluation. In deference to this prudent recommendation, the revised draft encompasses augmented empirical comparisons encompassing contemporary graph neural network (GNN) architectures, including CGCNN, Roost, and CrabNet. This comprehensive selection of apposite baselines will effectively validate the efficacy of our proposed approach.

### __Rationale for Baseline Selection: Stems from the structure-agnostic nature of the problem at hand.__

Unlike many Graph Neural Network (GNN) works that rely on incorporating extensive structural information into model inputs to enhance performance, our focus lies in scenarios where such structural information is unavailable. As outlined in the introduction, our real-world materials screening involves the exploration of conceptually designed novel materials with composition dopings, often lacking reliable structural data, such as precise atom positions. Models heavily dependent on detailed structural information would encounter significant performance challenges in these cases. Therefore, we deliberately selected structure-agnostic models, specifically Roost and CrabNet, as our baselines. For CGCNN, we made necessary adaptations to ensure its applicability in our problem domain, where 3D structural data are absent.

### __Inclusion of DFT-Generated Dataset: One more DFT dataset was selected for the sake of structural data and more material properties.__

The experimental datasets curated and utilized in the original manuscript were chosen due to the heightened reliability of experimental data compared to calculations, especially when employing diverse approximation methods. Additionally, experimentally measuring the properties of materials with low concentrations of dopings is more accessible. It’s useful in guiding our experimental designs of materials. However, a limitation arose from the absence of structural data for these materials and a constraint on available material properties.
To enable a more rigorous assessment, an additional DFT-generated hybrid perovskite dataset has been incorporated (in material science, DFT-generated data can be regarded as labeled data), encompassing an augmented range of quantifiable traits supplemented by precise structural representations. However, combinations of distinct constituents within the crystalline lattice sites are not included, probably due to the complexity of simulating the mixed crystals with DFT. The canonical CGCNN architecture, intrinsically equipped to harness structural data, has been appended to the revised evaluations on this dataset to facilitate equitable comparisons.

---

> ### Author Response · Authors · 2023-11-22
> **Suggestion on the Inclusion of Additional Baselines and Datasets #2**
>
> ### Results
>
> **Note that the training parameters were set as default; all models were trained 300 epochs and compared. There are two notable things from the results:**
>
> |        Datasets       | Curated |   DFT   |         DFT        |         DFT         |
> |:---------------------:|:-------:|:-------:|:------------------:|:-------------------:|
> |  Material Properties  | Bandgap | Bandgap | Atomization Energy | Dielectric Constant |
> |          SVR          |  0.0608 |    /    |          /         |          /          |
> |       **HP$^3$-NS**       |  **0.0483** |  **0.336**  |       **0.0150**       |         **3.40**        |
> |         Roost         |  0.0622 |  0.343  |       0.0164       |         3.30        |
> |        CrabNET        |  0.0515 |  0.373  |       0.0220       |         3.48        |
> |         CGCNN         |  0.0541 |  0.352  |       0.0185       |         3.34        |
> | CGCNN(with structure) |   N/A   |  0.141  |       0.0090       |         3.34        |
>
> **Our proposed method outperforms in hybrid organic-inorganic materials systems**: In the Curated Experimental dataset, where the material entries vary both in the organic molecules of crystal sites and the compositions, the role of better organic molecule embedding is significant. Notably, Roost and CrabNet, lacking differentiation between intra-organic molecule interactions and crystalline interactions, exhibit inferior performance compared to our methods. This underscores the significant contribution of our approach, which excels in capturing the nuanced interactions within these complex materials.
>
> In the DFT dataset, the role of organic molecules is more categorical since the dataset only contains their binary existence. So, the advantage of our method is relatively limited. But it still outperforms both structure-agnostic methods.
>
> **The CGCNN that uses structural data outperforms structure-agnostic methods**: It is an established fact that materials' properties are intricately tied to their structures. However, as elucidated in the introduction, structural data is often absent in conceptually designed new materials, especially when compositions are altered. We aimed to develop a robust method capable of representing complex material systems by leveraging the maximum available information about the structure. This approach facilitates large-scale and rapid screening of material properties, making it particularly advantageous in scenarios where structural data may be incomplete or unavailable, with reasonable compromise of accuracy.
>
> We sincerely hope the additions of pertinent baselines and datasets substantiate our method's efficacy and broad applicability. Please advise if any other avenues may be pursued to further strengthen the technical merits and presentation.

---

### Author Response · Authors · 2023-11-22
**General Reply: Suggestion on the Ablation Study**

We acknowledge the importance of conducting an ablation study to analyze the impact of individual components on the predictive performance of our proposed method. The revised manuscript will include a dedicated section for the ablation study, providing a comprehensive analysis of each model component. The results are summarized below, where models without subgraphs, without intra-group ratios, and without node weights were trained and evaluated to validate the improvements these designs introduced to the model.

|           Models           | Bandgap MAE |
|:--------------------------:|:-----------:|
|          **HP3-NS**           |    **0.0483**   |
|    Without nested graph    |    0.0522   |
| Without intra-group ratios |    0.0534   |
|    Without node weights    |    0.0499   |

The contributions of the components, presented in descending order, are outlined as follows:

### **Intra-group ratios: Encapsulating information of intra-lattice interaction probabilities**
The pivotal role of intra-group ratios lies in their integral contribution to the edge design within our structure-agnostic graph representation. In the context of constrained material crystalline systems, the edges between nodes must encapsulate information about interactions among these nodes. Within crystals, this information is best represented by the statistical presence in lattice sites. Consequently, the intra-group ratio assumes a critical role in property predictions, particularly in scenarios involving low concentrations of a component being doped into the crystal.

### **Nested graph: Differentiating handling of organic molecules with atomic sites**
The nested graph component corresponds to the heterogeneous embedding of organic molecules within crystalline structures. The differentiation in handling organic molecules and atoms theoretically enables a more accurate embedding of their roles in crystal structures. Given that our dataset predominantly comprises materials with organic dopings, the significance of nested graphs becomes particularly pronounced.

### **Node weights: Emphasizing the importance of more prevalent atoms or molecules**
Node weights are the ratios of interactions among nodes at the crystal level. They serve to emphasize the importance of more prevalent atoms or molecules when updating a node's features. While the introduction of similar information about statistical mixing has been addressed by intra-group ratios, the contribution of node weights is relatively limited. Nonetheless, it still enriches the model by incorporating additional knowledge about the constrained ABX3 structure and the relative importance among A, B, and X sites within the crystal.

We believe that addressing these concerns will significantly enhance our work's clarity, reliability, and applicability. We are committed to delivering a revised manuscript that reflects these improvements and provides a more robust contribution to the field.